# Neural cells are susceptible to historic and recently emerged Oropouche virus strains

Kaleigh A. Connors[1,2], Maris R. Pedlow[1,2], Zachary D. Frey[2], Marjorie Cornejo Pontelli[3], Sean P. J. Whelan[3], W. Paul Duprex[2], Leonardo D'Aiuto[4], Zachary P. Wills[5], Amy L. Hartman [1,2]*

1 Department of Infectious Diseases and Microbiology, School of Public Health, University of Pittsburgh, Pittsburgh, Pennsylvania, United States of America, 2 Center for Vaccine Research, University of Pittsburgh, Pittsburgh, Pennsylvania, United States of America, 3 Department of Molecular Microbiology, Washington University in St. Louis, St. Louis, Missouri, United States of America, 4 Department of Psychiatry, University of Pittsburgh, Pittsburgh, Pennsylvania, United States of America, 5 Department of Neurobiology, University of Pittsburgh, Pittsburgh, Pennsylvania, United States of America

* hartman2@pitt.edu

## Abstract

Oropouche fever is a re-emerging global viral threat caused by infection with Oropouche virus (OROV). While disease is generally self-limiting, historical and recent reports of neurologic involvement highlight the importance of understanding the neuropathogenesis of OROV. In this study, we characterize viral replication kinetics in neurons, microglia, and astrocytes derived from immortalized, primary, and induced pluripotent stem cell-derived cells, which are all permissive to infection with the prototypic OROV BeAn19991. We demonstrate cell-type dependent replication kinetics with both historic and recently emerged viral strains. Further, we show that ex vivo rat brain slice cultures can be infected by all OROV strains and produce antiviral cytokines and chemokines, which introduces an additional model to study OROV kinetics and tropism in the central nervous system. These findings provide insight into OROV neuropathogenesis and an initial assessment of newly emerged strains.

## Author summary

Oropouche fever is a recently re-emerging disease caused by infection with Oropouche virus (OROV). Historically, OROV has been overlooked and under-reported, and human disease was thought to be self-limiting. However, 2024 saw an unprecedented increase in cases and spread beyond Brazil, including an increased incidence of neurological disease in confirmed patients. Remarkably, very little is known about permissivity and cellular tropism of OROV for cells in the central nervous system. In this study, we characterize OROV viral replication kinetics in central nervous system cell types, including immortalized, primary, and induced pluripotent stem cell-derived cells. A comparison between a historical

**Data availability statement:** All relevant data are within the manuscript and supporting information metadata files.

**Funding:** This work was supported by the following National Institute of Health (NIH) and National Institute of Allergy and Infectious Diseases (NIAID) awards: R01 AI178378, R01 AI169850 and R56 AI171920 to ALH. This work was also partially supported by NIH/NIAID award U19 AI181984 (PI SPJW) to ALH and UC7 AI180311 (PI WPD) to support the Operations of the University of Pittsburgh RBL within the Center for Vaccine Research. The funders had no role in study design, data collection and analysis, decision to publish, or preparation of the manuscript.

**Competing interests:** The authors have declared that no competing interests exists.

OROV strain and emerging isolates provides an initial characterization of neuro-pathogenesis.

## Introduction

Oropouche virus (OROV) is a newly re-emerging arbovirus endemic to South America [1]. Belonging to the *Peribunyaviridae* family, OROV is transmitted to humans primarily through the bite of the *Culicoides paraensis* midge, and potentially mosquitos such as *Culex quinquefasciatus*, *Aedes aegypti*, and *Ochlerotatus serratus* [2,3]. Since its first identification in Trinidad in 1955, OROV has infected tens of thousands of individuals primarily in South America [4,5], where it has remained largely under the radar compared to the more well-known Dengue, Chikungunya, and Zika viruses. The disease Oropouche fever is a febrile illness characterized by severe headache, chills, arthralgia, myalgia, and maculopapular rash. Infrequently, cases of hemorrhagic fever and neurological involvement have been noted [2,6,7], and until recently, human deaths have not been associated with OROV infection.

At the end of 2023, larger outbreaks of OROV began in both endemic and new areas of South America [8]. Cuba reported its first ever locally acquired case in June of 2024, and there have been several travel-associated cases in the United States and Europe [9,10]. As of February 2025, over 16,000 confirmed cases of Oropouche fever were reported in Central and South America, the United States, and Canada, with Brazil reporting the highest number of cases [11]. Most concerningly, this recent outbreak has seen higher levels of apparent severe disease, including neurological issues such as the development of Guillain-Barré Syndrome (GBS) [12,13], and the first ever reported deaths due to hemorrhagic manifestations of OROV disease in Brazil [14]. Cases of mother-to-child transmission, resulting in microcephaly and even fetal death, have been reported for the first time [15]. Evidence indicates that the currently circulating strain is a result of a reassortment event involving the M segment from a previously circulating Oropouche virus strain in North Brazil and the S and L segments from a circulating strain in Peru, Colombia, and Ecuador [8,16]. Yet still, little is known about OROV neuroinvasive disease, which is estimated to occur in about 4% of cases [6,12,17,18]. Neurological symptoms, including dizziness, photophobia, confusion, and nystagmus, raise concerns about the ability of the virus to breach the blood-brain barrier and cause direct neuronal damage. The mechanisms by which OROV may invade the CNS are not well understood, hindering the development of effective therapeutic interventions.

To better understand the potential neuropathogenesis of OROV, we characterized OROV infection in several model systems, including immortalized human and rodent cell lines, primary cells, human induced pluripotent stem cell-derived (hiPSC-derived) neurons, and ex vivo rat brain slice cultures (BSCs). Neural progenitor cells (NPCs), neurons, astrocytes, and microglia from different species were all permissive to OROV infection, resulting in virus amplification. Additionally, primary cortical neurons supported the replication of both historical and newly emerged OROV strains.

We translated these findings to an ex vivo neural model and show that multiple OROV strains replicate in BSCs and induce changes in pro-inflammatory genes. These findings demonstrate that historical and emergent OROV strains can infect and replicate in various neural cell types and model systems, which is crucial for furthering our understanding of strain-specific pathogenesis of emerging viruses.

## Results

### Oropouche virus replicates in neurons, microglia, and astrocytes

Prior to the 2024 outbreak, we assessed OROV-BeAn19991 replication kinetics in CNS-related immortalized cell lines derived from mice and humans. We used proliferating mouse N2a and human neuroblastoma SH-SY5Y cells as a proxy for neurons. The N2a cells were infected with OROV-BeAn19991 at an MOI of 0.1 or 0.01. Infection resulted in a 3-log increase in viral RNA titers (as measured by plaque forming unit equivalents/mL; pfu eq./mL) by 72 hours post infection (hpi) (Fig 1A). Due to an increase in seeding density, we then infected with OROV-BeAn19991 at an MOI of 0.01 or 0.001 in the SH-SY5Y cells, where we also observed a 3-log increase in viral titers by 72 hpi (Fig 1B). Viral titers reached 1 x $10^7$ pfu eq./mL and 1 x $10^{6.5}$ pfu eq./mL by 72 hpi in each neuronal cell line, respectively. Immunofluorescent microscopy of viral antigen at 48 hpi show substantial staining of both N2a and Sh-SY5Y cells alongside antibodies against Nestin (neural progenitor cells) and Beta III tubulin (neuronal differentiation) (Fig 1G).

We then assessed OROV-BeAn19991 replication kinetics in microglial cell lines. Both BV2 (mouse) cells and HMC-3 (human) cells were infected with OROV-BeAn19991 at an MOI of 0.1 or 0.01. Viral RNA titers increased by 3-logs in BV2 cells and 2-logs in HMC-3 cells by 72 hpi (Fig 1C and 1D). In BV2 and HMC-3 cells, viral titers reached 1 x $10^6$ pfu eq./mL and 1 x $10^{5.5}$ pfu eq./mL, respectively. Immunofluorescent microscopy of BV2 and HMC-3 cells demonstrates widespread viral antigen staining by 48 hpi (Fig 1H).

We next looked at OROV-BeAn19991 replication in astroglial cells. We used the type 1 astrocyte cells C8-D1A (mouse) and hTERT-immortalized human astrocytes infected with an MOI of 0.01 or 0.001. Infection resulted in a 2-log increase in viral RNA titer by 72 hpi in C8-D1A cells (Fig 1E), and in a 1-log increase in hTERT-immortalized human astrocytes (Fig 1F). RNA titers in astrocytes reached 1 x $10^5$ pfu eq./mL and 1 x $10^{4.5}$ pfu eq./mL at 72 hpi, respectively. Viral antigen was detected by immunofluorescent microscopy in C8-D1A and hTERT-immortalized astrocytes at 72 hpi, consistent with peak viral RNA titers (Fig 1I).

Taken together, these findings demonstrate that immortalized neurons, microglia, and astrocytes are all highly permissive to OROV in a dose-dependent and cell-type dependent manner, with apparently higher preference for neurons and microglia compared to astrocytes.

### Primary neural progenitors and neurons are permissive to Oropouche virus infection

Subsequently, we assessed OROV replication kinetics in human induced pluripotent stem cell-derived (hiPSC-derived) neuroprogenitor cells (NPCs) and neurons, as well as primary rat cortical neuron cultures. Primary neuronal cells more accurately represent the morphological and physiological state of cells in vivo. Further, hiPSC-derived NPC cultures retain the ability to differentiate into various cell types, allowing for comparison of viral permissivity between immature and mature neuronal states [19]. A dose-dependent replication was observed in hiPSC-derived NPC and neuron cultures infected with OROV-BeAn19991 over 72 hpi, with a 2-log increase in viral titers in NPCs at 48 hpi (Fig 2A), and 1.5-log increase in neuronal culture by 72 hpi (Fig 2C). In NPCs, OROV-BeAn19991 replicates with an immediate increase in vRNA and antigen staining by 16 hpi (Fig 2A and 2B). In the neuronal cultures, there was an apparent lag time where increases in viral titer were not detected until 48 hpi. This can be observed using immunofluorescent microscopy, where the viral antigen staining of NPCs at 16 hpi is diffuse (Fig 2B), while in the neuron culture there are only a few positive cells observed until later time points (Fig 2D).

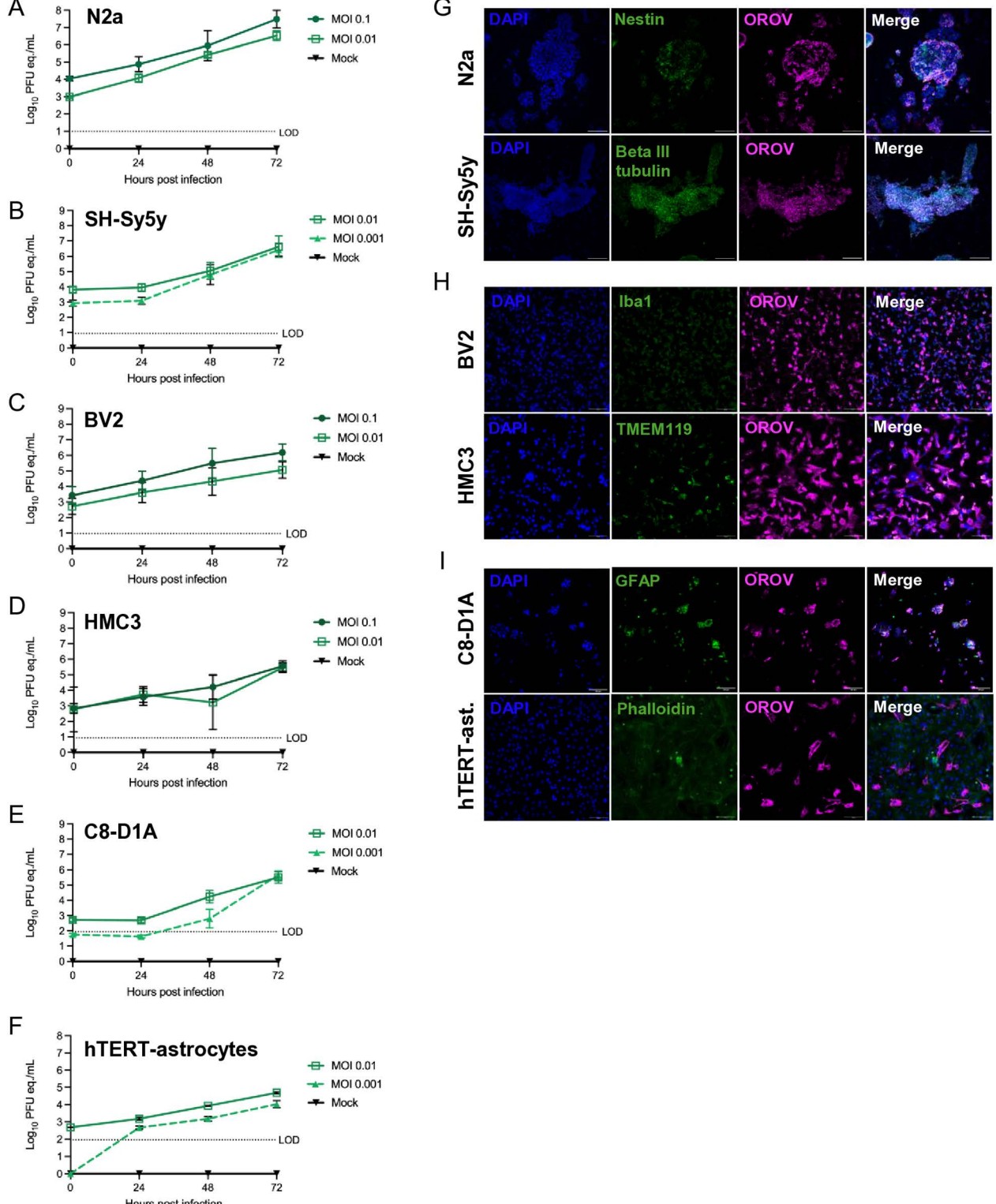

**Fig 1. Oropouche virus infects and replicates in neural cell lines in a dose-dependent manner.** Neuronal (A, B, G), microglial (C, D, H) and astrocyte (E, F, I) infection with OROV-BeAn19991. **(A)** Murine N2a cells, **(B)** human SH-SY5Y, **(C)** murine BV2, **(D)** human HMC-3, **(E)** murine C8-D1A, or **(F)** human hTERT-immortalized astrocyte cells were inoculated with OROV at MOI 0.1, 0.01, or 0.001 and viral RNA was quantified at 24, 48 and 72

hpi by RT-qPCR. The data for each cell line comprises two separate experiments. **(G)** N2a and SH-SY5Y cells were fixed at 48 hpi and immunostained for anti-OROV (magenta), anti-Nestin (green) or anti-beta III tubulin (green) and counterstained with DAPI (blue). **(H)** BV2 and HMC-3 cells were fixed at 48 hpi and immunostained for anti-OROV (magenta), anti-IBA-1 (green) or TMEM119 (green), and counterstained with DAPI (blue). **(I)** C1-D8A and hTERT-immortalized astrocytes were fixed at 72 hpi and immunostained for anti-OROV (magenta), GFAP or phalloidin (green) and counterstained with DAPI (blue). Slides were imaged at 20x magnification. Scale bar = 100 mm.

## Historical and emergent Oropouche virus strains replicate to different titers in primary neurons

During the course of this work, we obtained two emergent strains of OROV, designated as OROV-240023 and OROV-AM0088. Sequence alignment of the available sequences to the historic prototype strain BeAn19991 (herein designated OROV-BeAn19991) show very little amino acid divergence across the L, M and S segments (Table 1). The OROV-AM0088 virus used here was recovered using reverse genetics and is based on a sequence from a febrile patient in Brazil in 2024 [16]. A second strain, OROV-240023, is derived from a primary isolate from a febrile patient who had recently traveled to Cuba [20]. Amino acid (AA) sequences for each strain were aligned using NCBI BLAST and prototypic strain OROV-BeAn19991 was compared to OROV-AM0088 and OROV-240023. Comparing OROV-BeAn19991 to the 2023–2024 strains, in the L segment, ~2.5% AA divergence; for the M segment, ~1.6% AA divergence, and less than 1% AA divergence for the nucleoprotein region of the S segment. The small non-structural (NSs) protein for 240023 was not available in the NCBI database, but the NSs divergence between OROV-BeAn19991 and OROV-AM0088 was ~1%.

To compare the infectivity of these newly emergent strains to the historical prototype strain, we infected primary rat neurons with MOI 0.1 of OROV-BeAn19991, OROV-AM0088, or OROV-240023 and collected supernatants over 72 hpi. Infection in primary rat cortical neurons resulted in infectious titers up to $1 \times 10^5$ pfu/mL by 48 hpi (Fig 3A). At 48 hpi, OROV-BeAn19991 replicates to higher titer than OROV-240023, and by 72 hpi, OROV-BeAn19991 is significantly higher than OROV-AM0088. Microscopy of neurons infected at MOI 0.1 revealed diffuse viral antigen (OROV-N) staining at 24 hpi across all three strains (Fig 3B). These results demonstrate that primary neurons and NPCs from different species are highly permissive to OROV infection, where the virus replicates to high titers. In addition, we demonstrate that primary rat neurons are susceptible to the emergent strains OROV-AM0088 and OROV-240023, albeit at slightly lower levels than the BeAn19991 strain.

## Oropouche virus infection in brain slice culture induces pro-inflammatory cytokines

Ex vivo brain slice culture (BSC) preserves structural integrity of neurons and resident cells of the CNS, including astrocytes and microglia, which play pivotal roles in viral infection. We previously established and characterized rat BSC as a model for Rift Valley fever virus (RVFV), a related neurovirulent bunyavirus [21]. Using this same model, we characterized OROV growth kinetics and immune response to infection across the three strains of OROV. To promote infection within the BSCs, we opted for a high inoculum dose of $1 \times 10^5$ pfu, based on previous studies [21–23]. Individual coronal slices from postnatal day 6 rats were inoculated with $1 \times 10^5$ pfu OROV-BeAn19991, OROV-240023, or OROV-AM0088, and supernatant was collected up to 72 hpi. Infectious viral titer, measured by plaque assay, indicated that ex vivo rat BSC supported moderate viral replication over time (Fig 4A), with peak titers reaching $1 \times 10^3$ to $1 \times 10^4$ pfu/mL by 72 hpi. Using immunofluorescent microscopy, we observed an increase in viral antigen staining between 24 and 48 hpi across all three OROV strains (Fig 4B).

To get a basic understanding of the innate inflammatory response to OROV infection in BSC, we measured changes in select antiviral genes over time in homogenized BSC samples collected at 24, 48, and 72 hpi with OROV-BeAn19991, OROV-240023, or OROV-AM0088. OROV-BeAn19991 induced significant changes in expression of IFNβ at 24 and 48 hpi, and of IFNα at 48 hpi (Fig 4C and 4D). The expression of IFNβ and IFNα in BSCs infected with OROV-AM0088 and OROV-240023 were delayed until 48 hpi, and only OROV-240023 resulted in a significant increase in IFNα compared with mock-infected control slices at 48 and 72 hpi (Fig 4C and 4D). OROV-AM0088 induced significant changes in the

A

### hiPSC-derived NPCs

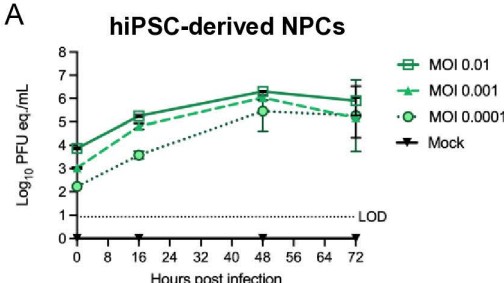

B

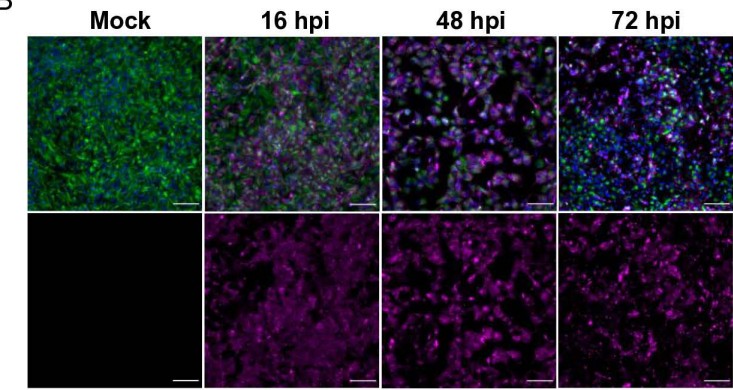

**DAPI Nestin OROV**

C

### hiPSC-derived neurons

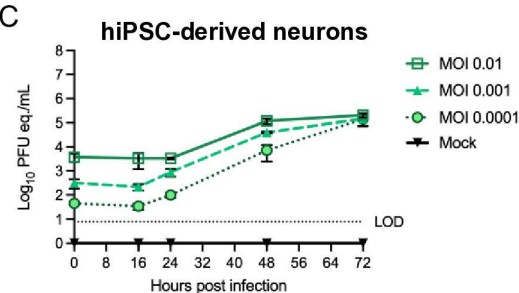

D

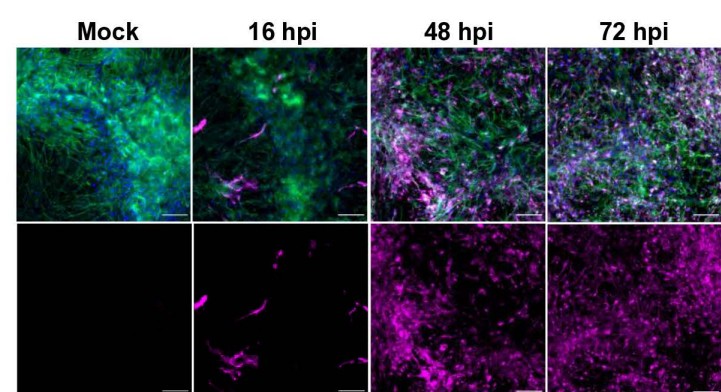

**DAPI Beta III tubulin OROV**

**Fig 2. Oropouche virus replicates to high titers in neural progenitor cells and neuronal cultures.** Human iPSC-derived NPCs **(A-B)** and neurons **(C-D)** were inoculated with OROV- BeAn19991 at MOI 0.01, 0.001, or 0.0001. Viral RNA was quantitated at 16, 24, 48, and 72 hpi by RT-qPCR. **(B)** hiPSC-derived NPCs and **(D)** hiPSC-derived neurons were fixed at 16, 48, and 72 hpi alongside mock-infected cells and immunostained with anti-OROV-N (magenta), anti-Nestin (green) or anti-beta III tubulin (green) and counterstained with DAPI (blue). Images obtained at 20x magnification. Scale bar = 100 mm.

**Table 1. Amino acid alignment of historic and emergent Oropouche strains.**

| Strain 1 | Strain 2 | Segment | Query length (aa) | No. of Differences | Amino acid Divergence (%) |
|---|---|---|---|---|---|
| BeAn19991 | AM0088 | L | 2197 | 54 | 2.45% |
| BeAn19991 | AM0088 | M | 1420 | 25 | 1.76% |
| BeAn19991 | AM0088 | S (NP) | 231 | 0 | 0% |
| BeAn19991 | AM0088 | S (NSs) | 91 | 1 | 1.01% |
| BeAn19991 | 240023 | L | 2252 | 55 | 2.44% |
| BeAn19991 | 240023 | M | 1420 | 22 | 1.54% |
| BeAn19991 | 240023 | S (NP) | 231 | 0 | 0% |

expression of IL-1β across all three timepoints sampled (Fig 4E). For all three strains, we detect significant increases in MCP-1 expression at 48 and 72 hpi (Fig 4F). These findings support the notion that historical and emergent OROV strains induce a pro-inflammatory response from resident neural cells within ex vivo BSCs in response to infection.

## Discussion

While OROV is primarily known for causing Oropouche fever, a self-limiting febrile illness, growing evidence suggests that the virus may have neuroinvasive properties, leading to CNS involvement [6,17]. For the first time, OROV infection has been associated with the development of GBS in adults [12]. Furthermore, OROV infection has been linked to miscarriage, stillbirths, and microcephaly in infants, which may previously have been reported but not further explored [15,24]. This neurological disease-causing aspect of OROV pathogenesis has been relatively understudied despite its potential implications for health of adults and neonates.

Studies in Golden Syrian hamsters and neonatal mice indicate both neurons and microglia are susceptible to OROV infection, but not astrocytes [25–27]. Here, we demonstrate that immortalized neurons, microglia, and astrocytes obtained from both rodent and human origin can be infected with OROV-BeAn19991 in vitro in a dose-dependent manner, although astrocytes were less permissive overall. Neurons supported the most viral replication, followed by microglia, then astrocytes. It is important to note that the neural cell lines (N2a and SH-SY5Y) used in this study were not differentiated prior to infection, and in general, immature neurons have a lower capacity for antiviral response than mature neurons [28,29]. Further, microglia and astrocytes are more immunoreactive and can induce a stronger antiviral immune response, potentially limiting OROV replication in the monolayer culture system used here [30]. The role that microglia and astrocytes play during OROV infection, including potential harmful effects of their activation, is an area in need of further study.

To move beyond transformed cell lines, we examined permissivity of primary rat neurons and human induced pluripotent stem cell (iPSC)-derived neural cells. Human NPCs, which are immature neuroprogenitor cells, were highly susceptible to OROV-BeAn19991 infection, even at low MOIs, compared to both more mature hiPSC-derived differentiated neuronal cultures and mature primary rat neurons. NPCs are found in the developing and adult nervous system, and can proliferate and differentiate into different types of neural cells, including neurons, astrocytes, and oligodendrocytes [31]. It is known that NPC infection by many viruses, including Zika virus (ZIKV), Japanese encephalitis virus (JEV), and Lymphocytic Choriomeningitis virus (LCMV) results in premature differentiation, reduced proliferation, dysregulation of cell cycle, and cell death [32–36]. While our studies suggest that perhaps undifferentiated neural cells are more permissive to OROV infection, the effects of OROV infection during neurogenesis, and the susceptibility of mature versus immature neurons, requires further evaluation.

The historical prototype strain of OROV used in most of these analyses, BeAn19991, was isolated from a sloth in Brazil in 1960 and falls within Lineage I [37]. This is also the strain used to develop the original OROV reverse genetics system in 2016 [38]. The 2024 outbreak in Brazil was the result of a recombination event leading to a new OROV lineage, BR-2015–2023,

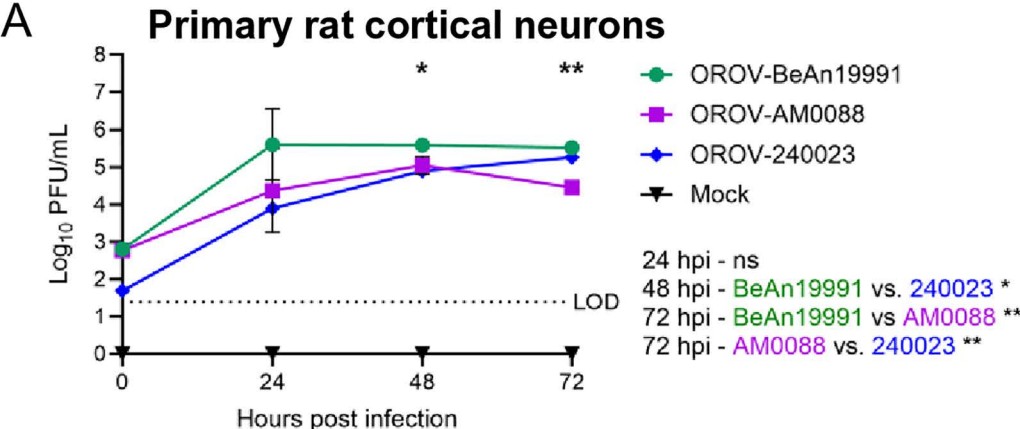

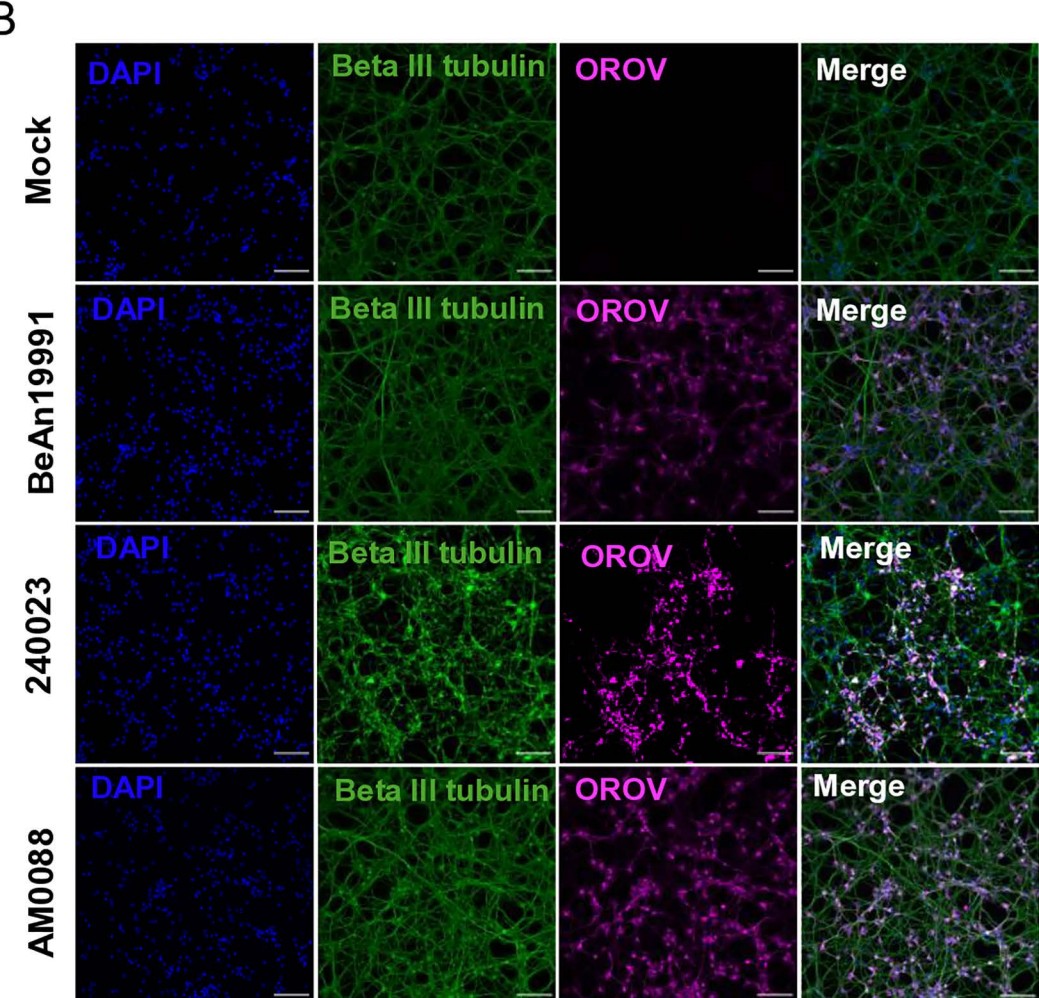

**Fig 3. Historical and emergent Oropouche virus strains replicate in primary rodent cortical neurons.** Primary rat cortical neurons were infected at 4 days in vitro with OROV-BeAn, OROV-240023, or OROV-AM0088 at MOI 0.1. **(A)** Viral plaque assay was used to quantitate infectious virus at 24, 48, and 72 hpi. **(B)** Primary rat cortical neurons were fixed at 24 hpi and immunostained with anti-OROV-N (magenta), anti-beta III tubulin (green), and counterstained with DAPI (blue). Images obtained at 20x magnification. Scale bar = 100 mm. Statistical analysis performed using a two-way ANOVA. *P < 0.05; **P < 0.01; ***P < 0.001; ****P < 0.0001; ns = not significant.

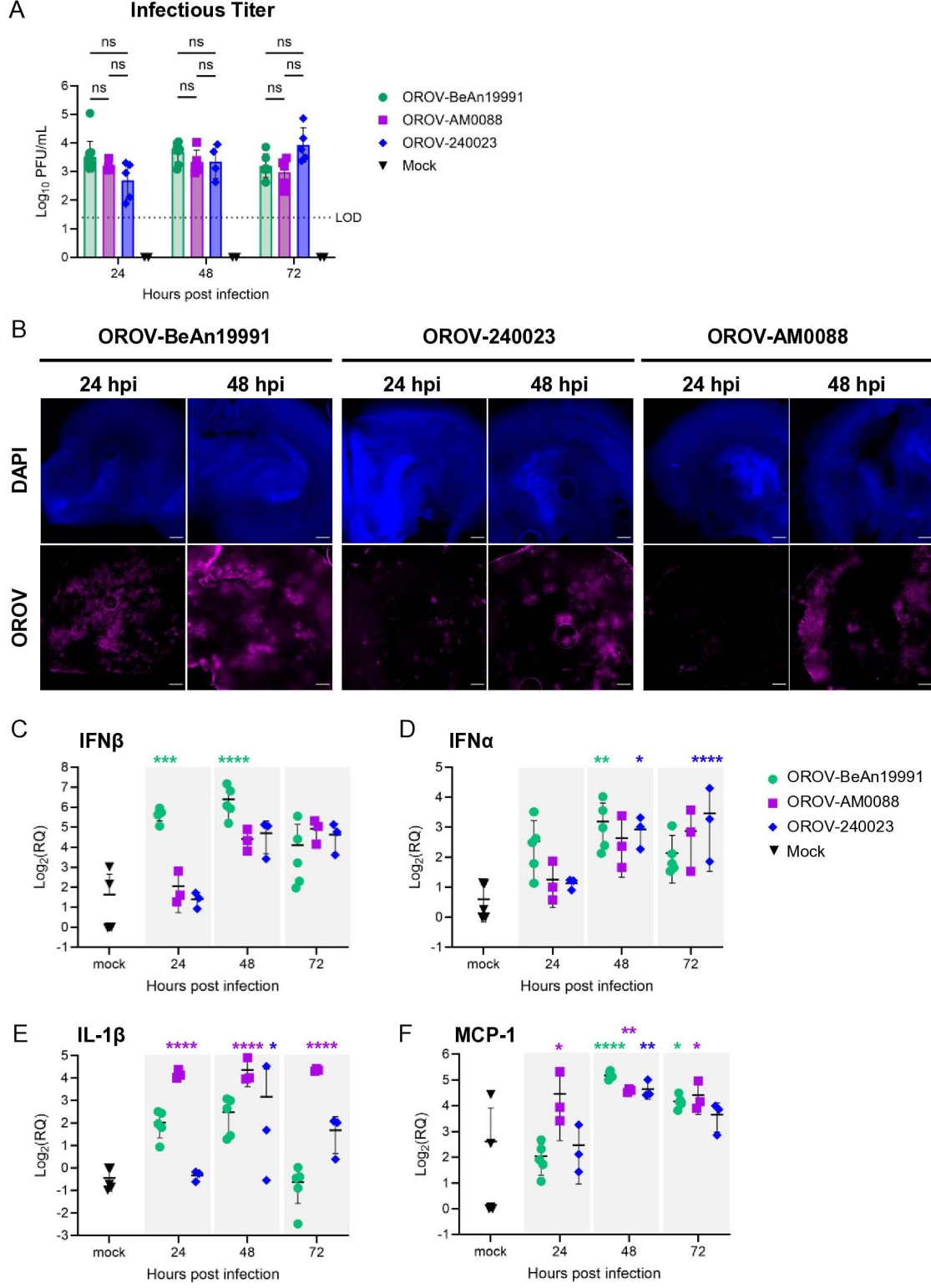

**Fig 4. Historical and newly emerged Oropouche virus strains infect and replicate in ex vivo brain slice cultures, resulting in antiviral gene expression. (A)** Coronal rat brain slices were inoculated with $1 \times 10^5$ pfu OROV- BeAn19991, OROV-240023, OROV-AM0088, or mock infected. Infectious titer from supernatants were quantitated using viral plaque assay at 24, 48, and 72 hpi. Infectious titer data is a combination of OROV-infected

brain slices (BeAn19991, n = 8; 240023 and AM0088, n = 5) and mock-infected brain slices (n = 6) across two independent experiments. **(B)** Whole BSCs were fixed at 24 and 48 hpi, and immunostained with anti-OROV-NP (magenta) and counterstained with DAPI (blue). Images of one hemisphere were obtained at 2.5x magnification. Scale bar = 500 mm. **(C-F)** Rat BSC were infected with 1 x 10$^5$ pfu OROV- BeAn19991, OROV-240023, or OROV-AM0088 (BeAn19991, n = 5; 240023 and AM0088, n = 3) or mock infected (n = 5), and RNA was isolated at 24, 48, or 72 hpi from whole slices. Mock data represents mock infected slices from each timepoint, and gene expression data was pooled. Gene expression was quantified by RT-qPCR using delta-delta Ct and normalized to the variation of the amount of beta-actin within each slice. Fold changes are relative to mock infected control slice. Error bars represent standard deviation. Statistical analysis performed using a two-way ANOVA. *$P < 0.05$; **$P < 0.01$; ***$P < 0.001$; ****$P < 0.0001$; ns = not significant.

which may have enhanced replication and immune evasion allowing it to emerge [16,39]. While most of our studies were done with BeAn19991, we were able to conduct several comparative studies. The OROV-240023 strain was isolated from a febrile patient in Italy with a history of travel to Cuba in July 2024 [20]. The AM0088 strain was isolated from sera of an Oropouche fever patient from Manaus City, Amazonas, Brazil [16], and the virus used here was recovered via reverse genetics from the published sequence. OROV-AM0088 and OROV-240023 are nearly identical at both the nucleotide and amino acid levels, indicating circulation of similar strains in Brazil and Cuba in 2024 [8,40]. Our analysis further supports a previous report demonstrating OROV-BeAn19991 and OROV-AM0088 share >97% amino acid sequence identity across all segments [16].

Once we had the recent strains of OROV, we compared replication of all 3 strains in primary rat cortical neurons and ex vivo brain slice cultures. Primary rat neurons were readily infected by all strains of OROV, although OROV-BeAn19991 may replicate faster than the 2023–2024 outbreak strains in this system. This is in contrast to recent reports that the 2023–2024 isolates have higher capacity for replication in Vero E6, Huh7, U-251, and W8 midge cells [16,41]. The reasons for these differing observations are unknown, but OROV-BeAn19991 has a longer passage history and thus may be cell culture or rodent adapted compared to the recently isolated strains which have not been passaged in vitro. Additional studies are needed to determine the factors driving differences in replication capacity across cell types.

Compared to immortalized cell lines or even primary neurons, ex vivo BSCs offer a multidimensional tool to understand neural susceptibility to viral infection and local immune response to viral infection through maintenance of cytoarchitecture and neural connections [42,43]. A prior study found that human brain slices were susceptible to OROV infection, with a preference for microglia, followed by neurons, although there was no astrocyte infection [27]. This is supported with an in vivo study that indicates astrocytes are activated during OROV infection of neonatal mice without noting infection of astrocyte cells [26]. Here, we show that rat BSCs are susceptible to multiple OROV strains resulting in modest levels of infectious virus production with substantial antigen spread throughout the tissue over time. The two emergent OROV strains were delayed in replication in BSCs, which is similar to what we saw in the primary neurons.

Following infection with OROV, we were interested in elucidating the cytokine response produced in the BSC tissue slices. All strains induced production of one or more of the following: IFN-α, IFN-β, IL-1β, and MCP-1. There were notable differences between strains and across timepoints. OROV-BeAn19991 induced higher expression levels of IFN-α and IFN-β compared with OROV-240023 and OROV-AM0088, which may explain the reduction in viral titer in BeAn19991-infected BSCs by 72 hpi. The delayed IFN-α response with OROV-240023 and OROV-AM0088 could be related to the slower replicative capacity observed in our BSCs. Interestingly, IFN-α was found to be strongly induced during Oropouche fever patients in a 2019 study [44], and OROV is restricted in non-myeloid cells by IFN pathway-related signaling molecules MAVS, IRF-3, and IRF-7 [45]. Like other bunyaviruses, OROV encodes a non-structural protein (NSs) which interferes with host transcription and is an interferon antagonist [38]. Despite this, OROV readily induces a type I interferon response, which is recapitulated in this system [45,46]. The kinetics and longevity interferon response to OROV infection in the CNS, and the role of OROV-NSs in neurovirulence are goals of future studies.

OROV induced variable levels of IL-1β in BSCs. The release of IL-1β is associated with the cell death mechanism pyroptosis, which likely contributes to the observed cytopathic effects of OROV infection in vitro. A recent report indicates that infection with OROV strain MD023 triggers the NLRP3 inflammasome in THP-1 macrophages, releasing IL-1β [47].

All three OROV strains significantly induced MCP-1, a critical chemokine which attracts immune cells, including mono-cytes, macrophages, and T cells, to sites of infection. In the brain, MCP-1 activates microglia and astrocytes, resulting in further immune signaling and recruitment of immune cells from the periphery. The elevation of interferon genes, and high expression of IL-1β and MCP-1 during OROV infection, is in line with prior reports [44–46]. Overall, we observed signifi-cant increases in antiviral cytokine and chemokine expression, albeit to lower levels than was induced by RVFV in ex vivo rat BSCs under similar experimental conditions [21]. These studies highlight the need to better understand bunyaviral neu-ropathogenesis, including viral tropism and antiviral immune response, in rat BSCs.

The development of experimental models is needed to support emerging and re-emerging viruses, including for OROV. This report describes in vitro and ex vivo models for studying OROV pathogenesis in the CNS and the antiviral immune response to infection. Our results indicate the susceptibility of neural progenitor cells to OROV infection, which is critical to explore given the recent cases of vertical transmission and the described microcephaly. Finally, these studies provided evidence to support neurovirulence of both historical and emergent OROV strains, highlighting the need for further study of the neuropathogenesis of this bunyaviral infection.

## Materials and methods

### Ethics statement

All work with animals adhered to The Guide for the Care and Use of Laboratory Animals published by the NIH throughout the duration of the study. The University of Pittsburgh is fully accredited by the Association for Assessment and Accred-itation of Laboratory Animal Care (AAALAC). The University of Pittsburgh Institutional Animal Care and Use Committee (IACUC) oversaw this work and approved it under protocol numbers 21100165 and 22051190.

### Biosafety

Work with OROV was completed in a Biosafety Level 2 (BSL-2) laboratory following all university biosafety guidelines.

### Viruses

The BeAn19991 strain of OROV (OROV-BeAn19991) was rescued through reverse genetics and was kindly provided by Paul Duprex (Pitt Center for Vaccine Research) and Natasha Tilston-Lunel (Indiana University) (Accession numbers: KP052852.1, KP052851.1, KP052850.1) [38]. The 240023 strain (OROV-240023) was obtained from Brandy Russell of the Arbovirus Reference Collection, Centers for Disease Control and Prevention, Fort Collins, CO (Accession numbers: PQ417950.1, PQ417949.1, PQ417948.1).

The AM0088 strain (OROV-AM0088) was rescued by reverse genetics and provided by Sean Whelan and Marjorie Cornejo Pontelli (Washington University, St. Louis). The sequence of the OROV strain AM0088, isolated in the Amazon state of Brazil during the 2023–2024 outbreak [16] was kindly provided by Dr. Pritesh Lalwani. The sequence of this strain is available in GenBank as follows: large (PP992529), medium (PP992527), and small (PP992525) segments. To generate the recombinant AM0088 strain, we first created a consensus sequence for the UTRs by aligning the AM0088 genome with sequences from GenBank derived from isolates of the recent outbreak. Complete sequences for all three segments were then synthesized in antigenome orientation and cloned into rescue pTVT plasmids, following the method used for rescuing the BeAn19991 strain [38]. Briefly, pTVT plasmids encoding each antigenome segment were transfected into BSRT7/5 cells. After five days, the supernatant was collected, clarified, and used to produce the P1 stock. The P1 stock was amplified in Vero E6 cells to gener-ate the working stock, which was then deep sequenced to confirm it matched the reverse genetic genome sequence.

Viruses were propagated in Vero E6 cells with standard culture conditions using Dulbecco's modified Eagle's medium (DMEM) (ATCC, 30–2002) supplemented with 1% penicillin/streptomycin (Pen/Strep), 1% L-glutamine (l-Glut), and 2% (D2) fetal bovine serum (FBS). A standard viral plaque assay (VPA) was used to determine the titer of the stocks. The

agar overlay for the VPA was comprised of 2X minimal essential medium (MEM, ThermoFisher, 11935046), 2% FBS, 1% Pen/Strep, 1% HEPES buffer, and 0.8% SeaKem agarose (Lonza, BMA50010); the assay incubated at 37°C for 4 days, followed by visualization of plaques with 0.1% crystal violet.

## Cells

All BV2 cells (provided by Gaya Amarasinghe, Washington University, St. Louis) and Vero cells (American Type Culture Collection [ATCC], CRL-1586) were cultured in DMEM supplemented with 1% Pen/Strep, 1% L-Glut, and either 2% (D2), 10% (D10), or 12% (D12) FBS. SH-SY5Y (ATCC, CRL-2266) were cultured in D12/F12 media (ATCC, 30–2006) supplemented with 1% Pen/Strep and 1% L-Glut. N2a (provided by Gaya Amarasinghe, Washington University, St. Louis) and HMC-3 (ATCC, CRL-3304) cells were maintained in Eagle's minimum essential medium (EMEM) (ATCC, 30–2003) with 10% FBS and supplemented with 1% Pen/Strep and 1% L-Glut. C8-D1A (ATCC, CRL-2541) cells were maintained in DMEM supplemented with 10% FBS, 1% Pen/Strep and 1% L-Glut. Immortalized Human Astrocytes, fetal (hTERT) (hTERT-immortalized astrocytes) (ABM, T0281) were cultured in PriGrow IV (TM004) supplemented with 10% FBS, 1% L-Glut, 10ng/mL rhEGF (Z100139) and 1% Pen/Strep.

Prior to infection, N2a, SH-SY5Y, BV2, HMC-3, C8-D1A and hTERT-immortalized astrocytes cells were plated in 24-well plates, on poly-L-lysine (R&D Systems, 3438-100-01) coated coverslips, except for BV2 cells where poly-L-lysine was not used and hTERT-immortalized astrocytes cells, where extracellular matrix (ABM, G422) was used. Cells were plated at the following densities: N2a 100k-200k cells/well; Sh-Sy5y 500k-1 million (mil) cells/well; BV2 75k-100k cells/well; HMC-3 100k-200k cells/well; C8-D1A 200k cells/well; hTERT-immortalized astrocytes 175k cells/well.

## Isolation and culture of primary rat cortical neurons

On the day prior to neuron isolation, acid-washed coverslips were coated with PDL/Laminin (Sigma, P7405-5MG; Invitrogen, 23017–015). Dissociation media (DM) comprised of Hanks' Balanced Salt Solution (Invitrogen, 14175–103) supplemented with 10 mM anhydrous MgCl2 (Sigma, M8266), 10 mM HEPES (Sigma, H3375), and 1 mM kynurenic acid was prepared. DM was brought to a pH of 7.2 and sterile filtered prior to use. On the day of isolation, a trypsin solution containing a few crystals of cysteine (Sigma, C7352), 10 milliliter (mL) of DM, 4 microliter (µl) 1N NaOH, and 200 units of Papain (Worthington, LS003126); and a trypsin inhibitor solution containing 25 mL DM, 0.25g trypsin inhibitor (Fisher, NC9931428), and 10 µl 1N NaOH were prepared, and filter sterilized. Embryonic day 18 Long Evans (Crl:LE; Charles River Laboratories, Wilmington, MA) rats were dissected, and brains were removed. The cortices were separated from the hippocampus and placed into DM. 5 mL of trypsin solution was added and cortices were placed in a 37°C water bath for 4 min, swirling occasionally to mix. The trypsin solution was removed, and cortices were immediately washed with trypsin inhibitor once, and then twice more while swirling in the water bath. Following the washes, the trypsin inhibitor was removed and replaced with 5 mL of Neurobasal/B27 media, then triturated to dissociate the neurons. Final volume was brought to 10 mL of Neurobasal Plus/B27 Plus media, and cells were counted and plated at a density of 50k cells/well for 96-well plates, 100-150k cells/well for 24-well plates, or 2mil cells/well for 6-well plates. One hour after isolation, the media was removed and replaced with fresh Neurobasal/B27 media. Primary neuron cultures were maintained in Neurobasal/B27 media, which consists of standard Neurobasal Plus Medium (Gibco, A3582901) supplemented with 1% Pen/Strep, 1% L-Glut, and 2% B27 Plus Supplement (Gibco, A3582801).

## Generation of human iPSC-derived neural progenitor cells and neurons

Human NPCs were generated as previously described [48]. Briefly, hiPSCs were cultured in mTeSR1-plus medium supplemented with dual SMAD inhibitors SB 431542 and LDN 193189 to promote neural induction. After 8–10 days, neural rosettes were manually isolated, transferred into Matrigel coated plates and cultured in StemDiff Neural Progenitor Medium (STEMCELL Technologies, 05833) for the expansion of NPCs. All cells were cultured in standard conditions (37°C, 5% CO2, and 100% humidity).

NPCs were seeded into Matrigel-coated 12- or 6-well plates to the density of $2.5 \times 10^5$ or $5 \times 10^5$ cells/well, respectively, and cultured in neurobasal medium [Neurobasal medium supplemented with 0.5X B27 (Vitamin A+), 1% Pen/Strep, 1% Glutamax, BDNF (10 ng/mL), CHIR99021 (3 μM), Dorsomorphin (1 μM), Forskolin, and ROCK inhibitor (10 μM)]. Two days later, CHIR99021, Dorsomorphin, Forskolin (10 μM), and ROCK inhibitor were withdrawn and differentiating NPCs were cultured for 4 weeks. Half medium was changed every other day.

## Viral infection

Cells were plated in 12 or 24-well plates as indicated above prior to infection. Virus was diluted to indicated MOIs in 100–200 μl D2 media. Complete culture media was removed from wells and replaced with viral inoculum or mock (D2 only) infected. Cells were incubated for 1 hour at $37^oC$, 5% CO2 with rocking every 10–15 min. For immortalized cell lines, inoculum was removed and cells were washed once in 1X PBS, then cultured in D2 media. For primary neurons and iPSC-derived human NPCs and neurons, viral inoculum was removed and replaced with complete culture media. Supernatants were collected at indicated timepoints for vRNA in Trizol reagent (Invitrogen, 15596026) or frozen at $-80^oC$ for VPA.

## RT-qPCR

RNA isolation was performed with an Invitrogen PureLink RNA kit (Invitrogen, 12-183-025). Samples were inactivated 1:10 in Trizol reagent. 200 μl of chloroform was then added to each sample and allowed to sit at room temperature for three minutes. The samples were then centrifuged for 15 min at 12,000 x g at $4^oC$. After separation of the aqueous phase, the aqueous phase was removed, and an equivalent volume of 70% ethanol was added. The remainder of the isolation was performed with the PureLink RNA protocol. qRT-PCR was then performed using the Invitrogen SuperScript III Platinum One-Step Quantitative Kit (Invitrogen, 11732020). OROV primers targeting the S segment used were: OROV19991-Forward 5'-TACCCAGATGCGATCACCAA-3' and OROV19991-Reverse 5'-TTGCGTCACCATCATTCCAA-3'. The Taqman probe used was OROV19991-Probe 5'-6-FAM/TGCCTTTGGCTGAGGTAAAGGGCTG/BHQ_1–3'. Thermocycling parameters include the following: reverse transcription, $50^oC$ for 30 min; Taq polymerase inhibitor activation, $95^oC$ for 2 min; PCR amplification, $95^oC$ for 15 s; and $55^oC$ for 30 s (40 cycles). Semiquantitiation of virus was determined by comparing cycle threshold (CT) values from unknown samples to CT values from an in-house developed OROV BeAn19991 RNA standards based on PFU equivalents "PFU eq./mL". qRT-PCR was performed on the Quantstudio 6 (Applied Biosystems).

## Antibodies

The following primary antibodies were used for immunocytochemistry staining at 1:500 dilution: rabbit anti-OROV N (Custom Genescript), mouse anti-OROV serum (in house), anti-mouse Nestin (EMD-Millipore, MAB5326), anti-chicken Nestin (Novus Biologicals, NB100–1604), anti-chicken Beta III tubulin (EMD Millipore, AB9354), anti-rabbit Iba1 (FujiFilm, 019–19741), anti-rabbit TMEM119 (AbCam, ab210405), anti-chicken GFAP (AbCam, ab4674), conjugated Phalloidin (ThermoFisher, A12379). Secondary antibodies were used at 1:500 dilution: goat anti-mouse AF488 (Invitrogen, A11001), goat anti-mouse AF647 (Invitrogen, A21235), goat anti-rabbit AF594 (Invitrogen, A11012), goat anti-rabbit AF488 (Invitrogen, A11008), goat anti-chicken AF594 (Invitrogen, A32759), goat anti-chicken AF647 (Invitrogen, A32933TR), goat anti-mouse AF594 (Invitrogen, A11005) or goat anti-chicken AF488 (Invitrogen, A11039).

## Immunofluorescence

Upon harvest, virus-infected cells were fixed in 4% paraformaldehyde for 15 min, followed by 3 washes in 1X PBS. Coverglass was permeabilized with 0.1% Triton X-100 detergent in 1X PBS for 15 min at room temperature (RT). Cells were blocked using 5% normal goat serum (NGS) (ThermoFisher, 50062Z) for 1–3 hours (h) at RT, followed by incubation with primary antibodies for 1–2 h at RT. After 3 washes in 0.5X NGS, the secondary antibodies were added for 1 h at RT. The cells were counterstained with

Hoescht 33258 (Invitrogen, #H1398, 1:1000) and mounted using Gelvatol. Fluorescent slides were imaged using a Leica DMI8 inverted fluorescent microscope provided by the Center for Vaccine Research at 20X magnification or using a Nikon A1 confocal fluorescent microscope at the Center for Biologic Imaging. Images were processed and quantified using Fiji [49].

## Statistical analysis

Statistical analyses were performed using GraphPad Prism software (La Jolla, CA). For Fig 3A, a two-way analysis of variance (ANOVA) was performed. For Fig 4A and 4C–4F–4F, a two-way analysis of variance (ANOVA) was performed to compare results between OROV infected BSCs to mock infected controls across timepoints. Multiple comparison was performed using Dunnett's multiple comparison test. Significance indicated by: *, $P < 0.05$; **, $P < 0.01$; ***, $P < 0.001$; ****, $P < 0.0001$; ns, no significance.

## Supporting information

**S1 Table. Excel file of metadata for all figures.**
(XLSX)

## Acknowledgments

We thank Dr. Cynthia McMillen for her valuable insight and feedback on this paper.

## Author contributions

**Conceptualization:** Kaleigh A. Connors, Maris R. Pedlow, Zachary D. Frey, Sean P.J. Whelan, Leonardo D'Aiuto, Zachary P. Wills, Amy L. Hartman.

**Data curation:** Kaleigh A. Connors, Maris R. Pedlow, Zachary D. Frey, Sean P.J. Whelan, Leonardo D'Aiuto, Zachary P. Wills, Amy L. Hartman.

**Formal analysis:** Kaleigh A. Connors, Maris R. Pedlow, Zachary D. Frey, Leonardo D'Aiuto, Zachary P. Wills, Amy L. Hartman.

**Funding acquisition:** Amy L. Hartman.

**Investigation:** Kaleigh A. Connors, Maris R. Pedlow, Zachary D. Frey, Marjorie Cornejo Pontelli, W. Paul Duprex, Leonardo D'Aiuto, Zachary P. Wills, Amy L. Hartman.

**Methodology:** Kaleigh A. Connors, Maris R. Pedlow, Zachary D. Frey, Marjorie Cornejo Pontelli, Sean P.J. Whelan, W. Paul Duprex, Leonardo D'Aiuto, Zachary P. Wills, Amy L. Hartman.

**Project administration:** Amy L. Hartman.

**Supervision:** Amy L. Hartman.

**Visualization:** Kaleigh A. Connors, Amy L. Hartman.

**Writing – original draft:** Kaleigh A. Connors, Amy L. Hartman.

**Writing – review & editing:** Maris R. Pedlow, Zachary D. Frey, Marjorie Cornejo Pontelli, Leonardo D'Aiuto, Zachary P. Wills, Amy L. Hartman.

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
