## [Decision Letter · Decision Letter 0]

19 Nov 2025

Neural cells are susceptible to historic and recently emerged Oropouche orthobunyavirus strains

PLOS Pathogens

Dear Dr. Hartman,

Thank you for submitting your manuscript to PLOS Pathogens. After careful consideration, we feel that it has merit but does not fully meet PLOS Pathogens's publication criteria as it currently stands. Therefore, we invite you to submit a revised version of the manuscript that addresses the points raised during the review process.

We look forward to receiving your revised manuscript.

Kind regards,

Alison M Kell

Academic Editor

PLOS Pathogens

Thomas Hoenen

Section Editor

Editor-in-Chief

PLOS Pathogens

PLOS Pathogens

orcid.org/0000-0002-7699-2064

**Additional Editor Comments:**

All reviewers agree on the importance of understanding replication kinetics of OROV and the potential link to neuropathogenesis. However, concerns with methods descriptions that could limit reproducibility and the use of emergent strains in only a subset of experimental conditions prevents us from accepting this manuscript as presented. We encourage the authors to consider the critiques below before resubmission.

**Journal Requirements:**

At this stage, the following Authors/Authors require contributions: Amy L. Hartman. Please ensure that the full contributions of each author are acknowledged in the "Add/Edit/Remove Authors" section of our submission form.

3) We note that your Data Availability Statement is currently as follows: "All relevant data are within the manuscript.". Please confirm at this time whether or not your submission contains all raw data required to replicate the results of your study. Authors must share the “minimal data set” for their submission. PLOS defines the minimal data set to consist of the data required to replicate all study findings reported in the article, as well as related metadata and methods (https://journals.plos.org/plosone/s/data-availability#loc-minimal-data-set-definition).

4) Please amend your detailed Financial Disclosure statement. This is published with the article. It must therefore be completed in full sentences and contain the exact wording you wish to be published.

2) If any authors received a salary from any of your funders, please state which authors and which funders..

**Reviewers' Comments:**

Reviewer's Responses to Questions

**Part I - Summary**

Reviewer #1: The manuscript by Connors et al is a timely and important body of work on an increasingly important arbovirus pathogen. The authors characterize OROV replication kinetics in neurons, microglia and astrocytes derived from immortalized, primary, and induced pluripotent stem cell-derived cells. Essentially, they show cell-type dependent replication kinetics with both historic and recently emerged OROV strains. They also show that ex vivo rat brain slice cultures can be moderately infected by OROV and produce antiviral cytokines and chemokines. The work is significant and important to the scientific community, but some revision is required to improve, clarity, transparency and depth. Below are a few thoughts and comments for your consideration

Reviewer #2: The article by Connors et al reports a straightforward assessment of Oropouche orthobunyavirus (OROV) infectivity in a variety of cell models, using primarily a historical strain and then comparing, in a subset of the cell models, a pair of recently emergent strains. However, no differential infectious properties/ exacerbated infection was seen in the emergent strains which might be due to 1. not using more neurovirulent, replication-competent strains 2. not being able to identify differences in immune responses, cytopathic effects and/or viral entry mechanisms in the CNS. While the data are clearly presented and will be of interest to the virology community, they do not present any groundbreaking findings as described in the scope for a journal of this level. The paper would be more appropriate for a lower impact publication. Several suggestions are provided that the authors could consider before resubmission elsewhere.

Reviewer #3: In this manuscript, Connors et al. describe the growth profiles of Oropouche orthobunyavirus (OROV) (both historic and newly emerged strains) in microglia, astrocytes, neuronal primary cultures, cell lines, and human induced pluripotent stem cell derived cells (hiPSC-derived), which were all demonstrated to be permissive to OROV infection. In addition, they describe amino acid changes between the historic strain (OROV-BeAn19991) and recently isolated strains (OROV-AM0088 and OROV-240023) and characterize an ex vivo rat brain slice culture model for OROV infection.

**Part II – Major Issues: Key Experiments Required for Acceptance**

Reviewer #1: Line 326. While the BeAn19991 strain was derived from a reverse genetics system, the absence of detailing the passage history for this strain and the associated strengths and weaknesses that accompanies the passage history must be described.

The experimental design is really unclear why so many MOIs were used with just one OROV strain, and the recent isolates weren’t used for comparison in any of the immortalized neurons, microglia and astrocytes. This would be very informative information given that most research is focused around the recent isolates and not the 1960 isolate. The need for these comparisons are also supported by the differences observed between the ancestral and contemporary isolates in rat cortical neurons and BSC gene expression studies.

Reviewer #2: Major Comments:

1. Fig. 1G, H, I: the images lack sufficent resolution (e.g. beta III Tubulin staining is clustered, Iba1 staining is extremely faint, Phalloidin staining seems very odd and variable). The authors should I. Add scale bar II. Incorporate higher magnification, better quality images III. Add IFA quantification of viruses in different cultures IV. Stain astrocytes with classical markers like GFAP, GLAST etc.

2. The authors didn’t report any cytopathic effects (CPE) even after long endpoints e.g. 4 days. Is there no visible CPE even with a high MOI inoculum? If so, the authors should infect the cultures with more virulent lines like BR-2015-2023, to evaluate the cause of neurovirulence and lethality in emergent isolates.

3. In Fig. 2D, why do the neuronal primary cultures have only a few infected cells, even though neurons lack strong immune response and should be highly permissible to neuroviruses like OROV? The authors should use higher inoculum dose and emergent lethal isolates to test this.

4. In Fig. 3A, why are the kinetics of the new strains slower even though these are more emergent with neurological symptoms? Also, in Fig. 3B, 240023, the staining of beta III Tubulin seems more clustered, probably directing virus-induced trafficking/ modulation of host factors. The authors should further investigate such differential characteristics between historic and emergent strains. In addition, these observations should be correlated with CPE in a later time point/ higher MOI infection.

5. In Fig. 4, do the authors have any thoughts/explanation as to why the emergent neurovirulent strains produce similar PFUs/ml, compared to the historical strain? Also, the Panel 4B images look out-of-focus; the authors should provide clearer maximum-intensity projection images for slice cultures. The authors should parallelly stain with neuronal/ NPC markers to demonstrate cell-specific localization with virus (in a higher-magnification). In Panel C., the historical strain induces more IFNB compared to others, are the emergent strains capable of interfering the neuronal immune response, inducing neurovirulence?

6. The authors should compare the neuro-invasiveness in animal models to compare strain-dependent pathologies in vivo.

Reviewer #3: This paper represents a well-executed characterization of growth characteristics of OROV strains in a variety of permissive cell lines, primary cells, and ex vivo rat brain slice cultures. Overall, the growth kinetics are very similar between OROV-BeAn19991 and the two more recent isolates. However, the authors bury the lead and the most interesting difference between these strains appears to be the increased inflammasome induction by OROV-AM0088, which may have important consequences in the increased findings of OROV-associated neurologic disease. This finding is not highlighted in the abstract and comprises the tail end of the discussion—it should be a more prominent focus of the manuscript.

2. The reverse genetics system could be used to determine which sequence polymorphisms are associated with NLRP3 induction. Or, the authors could generate genome reassortants to determine which segment is influencing NLRP3 induction.

**Part III – Minor Issues: Editorial and Data Presentation Modifications**

Reviewer #1: Viruses section. Please provide some of the metadata for these viruses within the manuscript. Host, year of isolation, passage history, etc?

Line 52. OROV is the acronym for Oropouche virus. Please correct.

Lines 41-42 “including alarming incidences of neurological issues in confirmed patients.” is unclear. Please be more specific.

Lines 58-59 “…febrile illness with severe headache, chills, …” should be “…febrile illness characterized by headache, chills, …”.

Line 335. …Was kindly provided…

Lines 336-337. “To generate reverse genetics” is a very unclear phrase. Please rephrase for clarity.

Table 2 is missing the NSs comparison between BeAn19991 and 240023. Is this protein absent in the 240023 strain.

Lines 404-405. There was no attempt of description of the methods used to determine or confirm the differentiation into neurons. This should be performed for any cell type that was differentiated.

Figures 1 and 2 present results of infectious titers yet the figure legend and methods describe the use of RT-qPCR. No description is provided if they performed a standardization procedure to be able to graph RNA loads as pfu/ml. This is needed to assess the rigor of estimating your equivalents.

Line 146. “were” should be “where”

Line 167. Give the rational for using a high titer (10^5 pfu) for infecting the BSCs.

Lines 169-170. With the exception of OROV-240023, I’m not convinced of replication for the other 2 strains in BSCs since there is no or very little increase in viral titers. The authors don’t necessarily rule persistence of the virus over time in the culture. It would be good to know what the 0hr timepoint titer would be, to see if there was an increase between the 0hr and 24 hr time points.

Lines 204-210. This is a valuable discussion point to compare the sequence identities. However the authors don’t emphasize any particular discussion point, but instead restate some of the results. What does this sequence similarity mean in the context of the proposed altered pathogenesis of the recent emergent strains? Or is it important to compare sequence similarity among all the complete genomes among the current OROV sequences available to help support your point. This sort of general analysis could be cited if already done, or at least serve to increase citations of this work.

Line 222. This is unclear and needs rephrasing

Give some rational as to why the proliferating neurons (N2a and SHSY-5Y) were not differentiated but used as is for a proxy for neurons.

Lines 256-258. I think these comparisons should be included in this study. Maybe do a single strain with a reduced number of MOIs for comparison if needed. This comment was noted above as well.

Figure 4 and lines 289-296. It’s not a good comparison. If the gene expression studies in mock were done at all 3 time points, then shouldn’t there be 6 points and not 4 as seen in the graph? Also, is there another measure of cellular health you can use to rule out the fact that the increase IL-1B is result of virus cytopathic effects and not natural cell death during maintenance, which you will see in mock controls.

Reviewer #2: Minor Comments:

1. The authors should explain more about the difference between the historic and emergent strains of OROV (mainly on infectious capability, neurotropism, immune responses) in the introduction as that’s the main theme of this manuscript.

2. Do any prior studies demonstrate differences/ comparative pathologies between isolates from any severe infectious cases with neurological symptoms, any comparable data on those?

3. Fig. 1A-F should show different doses in different colors, the different shades of green are not easily distinguished.

4. In Fig. 1 C-D and G, compared to the neuronal lines, why is prominent virus infection observed by IFA that's not reflected in titer (i.e. reduced in microglia)?

5. Do the astrocytes maintain low plaques production to harbor the virus for a longer time i.e. act as a viral reservoir in neuroinfection?

6. In Fig. 2, the mocks should be put in left, followed by initial-to-distant time post infection. Also, Panel B and D lacks clarity (D: Mock b III Tubulin image is not focused), thus should be added with higher magnification images.

7. The authors should use L-Glutamine instead of I-Glutamine, in case they have not used something different.

8. The authors should mention the plasmids used for reverse genetics. This is an interesting system, so they may think of describing it in more detail.

9. The authors should use more neurovirulent strains like BR-2015-2023 (that they mention in discussion) to evaluate mechanisms of increased neuropathogenesis in emergent strains. Do the authors predict any potential difference between CNS entry mechanisms?

10. Considering the relatively modest amount of data presented, the discussion is extremely long and somewhat rambling. Would suggest presenting a more succinct and focused discussion.

Reviewer #3: Non

PLOS authors have the option to publish the peer review history of their article (what does this mean? ). If published, this will include your full peer review and any attached files.

**Do you want your identity to be public for this peer review?** For information about this choice, including consent withdrawal, please see our Privacy Policy .

Reviewer #1: No

Reviewer #2: No

Reviewer #3: No

**Figure resubmission:**

**Reproducibility:**



---

## [Decision Letter · Decision Letter 1]

26 Jan 2026

Dear Dr. Hartman,

We are pleased to inform you that your manuscript 'Neural cells are susceptible to historic and recently emerged Oropouche virus strains' has been provisionally accepted for publication in PLOS Pathogens.

Best regards,

Alison M Kell

Academic Editor

PLOS Pathogens

Thomas Hoenen

Section Editor

PLOS Pathogens

Sumita Bhaduri-McIntosh

Editor-in-Chief

PLOS Pathogens

orcid.org/0000-0003-2946-9497

Michael Malim

Editor-in-Chief

PLOS Pathogens

orcid.org/0000-0002-7699-2064

Reviewer Comments (if any, and for reference):

Reviewer's Responses to Questions

**Part I - Summary**

Reviewer #1: (No Response)

Reviewer #3: The strength of the revised manuscript is the characterization of the growth profiles of historic and newly emerged Oropouche orthobunyavirus in physiologically relevant cell lines (microglia, astrocytes, neuronal cultures etc.), which is important to understand the potentially evolving pathobiology of infections with this re-emerging virus. The revised manuscript addresses many of the reviewers' concerns and clarifies many technical points. However, it does not satisfy several reviewer concerns that were salient to understanding pathogenic differences between strains, including further exploration of differential inflammasome induction and neuorinvasiveness between strains (using animal models), thereby diminishing enthusiasm for the revised manuscript.

**Part II – Major Issues: Key Experiments Required for Acceptance**

Reviewer #1: (No Response)

Reviewer #3: While reverse genetics experiments to identify genetic changes associated with inflammasome induction have been deferred to later studies, an in vivo comparison of the neuroinvasive potential of the strains studied in an animal model (e.g. IFNAR-/- adult mice, weanling mice, or hamsters) should be included in this report.

**Part III – Minor Issues: Editorial and Data Presentation Modifications**

Reviewer #1: (No Response)

Reviewer #3: none. Figures are improved.

PLOS authors have the option to publish the peer review history of their article (what does this mean? ). If published, this will include your full peer review and any attached files.

**Do you want your identity to be public for this peer review?** For information about this choice, including consent withdrawal, please see our Privacy Policy .

Reviewer #1: No

Reviewer #3: No

---

## [Editor Report · Acceptance letter]

Dear Dr. Hartman,

We are delighted to inform you that your manuscript, "Neural cells are susceptible to historic and recently emerged Oropouche virus strains," has been formally accepted for publication in PLOS Pathogens.

Best regards,

Sumita Bhaduri-McIntosh

Editor-in-Chief

PLOS Pathogens

orcid.org/0000-0003-2946-9497

Michael Malim

Editor-in-Chief

PLOS Pathogens

orcid.org/0000-0002-7699-2064